# Pretreatment with IL-15 and IL-18 rescues natural killer cells from granzyme B-mediated apoptosis after cryopreservation

Abdulla Berjis [1,2] ✉, Deeksha Muthumani[1,2], Oscar A. Aguilar[3], Oz Pomp[4], Omar Johnson[1], Amanda V. Finck [1,5], Nils W. Engel[1], Linhui Chen[1,6], Nicolas Plachta [4], John Scholler[1], Lewis L. Lanier [3], Carl H. June [1,7,8] & Neil C. Sheppard [1,8] ✉

Human natural killer (NK) cell-based therapies are under assessment for treating various cancers, but cryopreservation reduces both the recovery and function of NK cells, thereby limiting their therapeutic feasibility. Using cryopreservation protocols optimized for T cells, here we find that ~75% of NK cells die within 24 h post-thaw, with the remaining cells displaying reduced cytotoxicity. Using CRISPR-Cas9 gene editing and confocal microscopy, we find that cryopreserved NK cells largely die via apoptosis initiated by leakage of granzyme B from cytotoxic vesicles. Pretreatment of NK cells with a combination of Interleukins-15 (IL-15) and IL-18 prior to cryopreservation improves NK cell recovery to ~90-100% and enables equal tumour control in a xenograft model of disseminated Raji cell lymphoma compared to non-cryopreserved NK cells. The mechanism of IL-15 and IL-18-induced protection incorporates two mechanisms: a transient reduction in intracellular granzyme B levels via degranulation, and the induction of antiapoptotic genes.

NK cells are innate immune cells that make up 5–15% of the total circulating human lymphocyte population[1], and function to recognize and eliminate host cells experiencing cellular stress or loss of human leukocyte antigen (HLA) due to infection or neoplastic transformation, as well as to orchestrate adaptive immune responses[2–5]. NK cells possess several attributes that make them promising candidates as off-the-shelf cell therapies for cancer. These include their ability to broadly recognize tumor cells, including those that have downregulated HLA to escape T cells, and their promising safety profile as they do not mediate graft–versus–host disease and pose a low risk of inducing cytokine release syndrome (CRS) or immune cell-associated

neurotoxicity syndrome (ICANS)[6]. Currently, there are numerous clinical trials of unmodified and gene-modified, NK cells in progress, including CD19-specific chimeric antigen receptor (CAR)-NK cells for lymphoma[7]. Moreover, since NK cells are potent mediators of antibody-dependent cellular cytotoxicity (ADCC), they can be readily redirected towards selected targets via combination with monoclonal antibodies (mAbs)[8,9], or bispecific molecules engaging CD16[10] or other NK activating receptors.

The eventual commercialization of NK cell therapy will depend upon the ability to provide potent cryopreserved formulations. However, unlike CAR-T cells, where non-cryopreserved or cryopreserved

[1]Center for Cellular Immunotherapies, University of Pennsylvania, Philadelphia, PA, USA. [2]School of Engineering and Applied Science, University of Pennsylvania, Philadelphia, PA, USA. [3]Department of Microbiology and Immunology and Parker Institute of Cancer Immunotherapy, University of California; San Francisco, San Francisco, CA, USA. [4]Department of Cell and Developmental Biology, Institute for Regenerative Medicine, Perelman School of Medicine, University of Pennsylvania, Philadelphia, PA, USA. [5]Perelman School of Medicine, University of Pennsylvania, Philadelphia, PA, USA. [6]Institute for Biomedical Informatics, the Bioinformatic Core, Perelman School of Medicine, University of Pennsylvania, Philadelphia, PA, USA. [7]Parker Institute for Cancer Immunotherapy, University of Pennsylvania, Philadelphia, PA, USA. [8]Department of Pathology and Laboratory Medicine, University of Pennsylvania, Philadelphia, PA, USA. ✉e-mail: Abdulla.Berjis@Pennmedicine.upenn.edu; neil.sheppard@pennmedicine.upenn.edu

cells appear to have equivalent efficacy[11], cryopreserved expanded and activated (EA) NK cells have profound functional defects, with decreased granzyme B (GZMB) expression and cytotoxicity[12,13] and impairments in their migratory capability compared to non-cryopreserved NK cells[14]. As testament to the difficulty in using cryopreserved NK cells, fewer than 15 of over 600 NK cell trials registered at clinicaltrials.gov mention cryopreservation[15], and only one trial provided a head-to-head comparison of non-cryopreserved versus cryopreserved EA NK cells[13]. In this study, the median viability of cryopreserved NK cells fell from 94% upon thaw to 16% after overnight incubation and the thawed NK cells were not cytotoxic in vitro unless pulsed with IL-2[13]. Importantly, the cryopreserved NK cells failed to expand in patients, whereas non-cryopreserved NK cells expanded a median of 21-fold[13]. Even trials utilizing cryopreserved EA NK cells have often provided an initial infusion of non-cryopreserved NK cells[16], making the activity of the cryopreserved component difficult to discern. While progress in NK cell cryomedia and conditions have been made, prior reports have either assessed NK cell viability and function immediately post thaw using 4 h cytotoxicity and cytokine production assays[12,17,18] or made such assessments following recovery in IL-2-supplemented media[19,20]. The first approach likely underestimates the defects caused by cryopreservation by assaying NK cell activity prior to the reported time of cell death[13], while the latter approach is hindered by a lack of GMP facilities at the bedside and is thus inconsistent with current clinical practice wherein cryopreserved cell therapy products are thawed at 37 °C and infused within 30 min[21]. Until now, the mechanism by which cryopreservation affects NK cells has never been reported, limiting opportunities to explore hypothesis-driven interventions.

Here, we use healthy donor NK cells to investigate the mechanism of NK cell death and dysfunction after cryopreservation, and explore rational interventions to enable equipotent cryopreserved NK cells. We find that cryopreserved NK cells die by apoptosis post thaw due to GZMB leakage from cytotoxic granules. Pretreating NK cells with IL-15 and IL-18 before cryopreservation reduces intracellular GZMB levels via degranulation, and significantly upregulates the antiapoptotic gene *BCL2L1*, resulting in cryopreserved NK cells with 90–100% recovery and equal potency compared to non-cryopreserved NK cells. As this facile solution relies on readily available reagents and requires no specialist cryomedia or equipment, it may be readily implemented to enhance the NK cell therapy field.

## Results

### Characterizing NK cell recovery, phenotype, and function after cryopreservation

First, we characterized the effects that cryopreservation has on the recovery and function of EA NK cells using GMP-compliant cryomedia (CryoStor 5® or 10®) and the research scale, controlled-rate CoolCell® cell freezer. We measured the viability of NK cells before and after cryopreservation (Fig. 1a) using flow cytometry with antibodies and viability dyes. Consistent with prior findings[12,13,17,18], we did not see drastic changes in NK cell viability, which was due to short-term assessments and experimental bias stemming from the use of flow cytometry gating strategies that exclude dead cells (Supplementary Fig. 1a). In a time course after thawing, we found a drastic loss of NK cell numbers by 24 h post-thaw compared to immediately after thaw (time 0 h) (Fig. 1b). Next, we assessed whether the NK cells that survived cryopreservation maintained their effector functions. We performed degranulation and cytotoxicity assays 24 h after thawing NK cells using either K562 or *B2M*$^{-/-}$ Raji cells (Supplementary Fig. 1b, c and Fig. 1c–e) as target cells. Surprisingly, cryopreserved cells had higher surface expression of the degranulation marker CD107a than non-cryopreserved NK cells when cultured with K562 cells (Fig. 1c), however, we found that cryopreserved NK cells were significantly less cytotoxic (Fig. 1d, e). We also investigated the ability of the non-

cryopreserved and cryopreserved NK cells to mediate ADCC against the CD20$^+$ mantle cell lymphoma (MCL) cell line Jeko-1 (Supplementary Fig. 1d). Cryopreserved NK cells showed a significant loss of ADCC compared to non-cryopreserved NK cells (Fig. 1f). We assessed the impact of cryopreservation on the single-cell cytokine secretome of NK cells in response to R848, an agonist of Toll-like receptors (TLRs) 7 and 8 (see ref. 22). Interestingly, cryopreserved NK cells had a higher Polyfunctional Strength Index (PSI), a product of the number of cells secreting more than one cytokine and the intensity of the signal for each secreted factor (Fig. 1g), which agrees with their increased CD107A expression (Fig. 1c).

Next, we investigated the effect of cryopreservation on the NK cell transcriptome at 0, 24, and 72 h post thaw. Via differential expression analysis we found 153 and 209 genes were differentially expressed at 24 and 72 h post thaw, respectively (*P*adj >0.05). Interestingly, we observed trends of downregulation of chemokine receptors and ligands (Fig. 1h), which concurs with reported findings of defects in NK cell motility post thaw[14]. Gene Set Enrichment Analysis (GSEA) revealed the IL-2-induced STAT5 pathway, which is critical for NK cell activation and survival[23], was upregulated post thaw (*P*adj $1.1 \times 10^{-5}$) (Fig. 1h). The RNA sequencing analysis, together with the degranulation and cytokines secretion data suggests either that the cryopreservation and thawing process activates NK cells or those that are activated become enriched through survival or proliferative benefits post thaw. Lastly, we tested an NK cell cryopreservation protocol using cryomedia that includes 40% serum, which has been reported to enhance NK cell recovery[20] but found EA NK cells from our expansion protocol showed no improvement in recovery using this serum-rich cryomedia compared to CryoStore 10 (Supplementary Fig. 1e).

In sum, our data concur with prior reports[12–14] of a substantial loss of NK cell numbers and cytotoxic function following cryopreservation. We further found that the NK cells that survive cryopreservation have enhanced secretory activity.

### GZMB leakage induces apoptosis in NK cells after cryopreservation

Cryoinjury largely results from the formation of ice crystals that disrupt cellular structures[24]. Gross structural damage to NK cells might cause necrosis, although this is unlikely given the use of DMSO as a cryoprotectant and prior data[13] as well as our own findings (Fig. 1a, b) suggesting that NK cells are not lysed by cryopreservation, but die later by an active process. Such an active process might be the induction of apoptosis either directly or via fratricide. To investigate, we measured NK cell numbers before cryopreservation and at time 0 h after thawing and found no significant cell loss immediately after thawing (Fig. 2a), ruling out immediate necrosis. Next, in a time-course assay, we measured caspase-3/7 activation and staining with a vital dye to assess early apoptosis and cell membrane integrity, respectively. Caspase-3/7 activation was observed in cryopreserved NK cells above the levels seen in non-cryopreserved NK cells from 4 h post thaw and rose rapidly before reaching a plateau after 9 h (Fig. 2b). After 12 h post thaw, we could detect vital dye uptake, indicative of cell death, which increased through the remainder of the time course (Fig. 2c). These experiments confirmed the expected scenario supported by prior evidence[13] that cryopreserved NK cells go through programmed cell death after thawing and are not necrotic due to acute physical injury.

Considering the stark differences in cryopreservation tolerance between CD8 + T cells and NK cells, despite both being cytotoxic lymphocytes, the greater granularity of NK cells due to high levels of preformed cytotoxic granules[25] and their unique ability to recognize cell stress[26] emerge as key points of differentiation. In this context, apoptosis in cryopreserved, thawed NK cells might conceivably be due to autolysis if cryoinjury results in intracellular GZMB leakage from preformed cytotoxic granules, or due to fratricide following the induction of cellular stress and its recognition by neighboring NK cells.

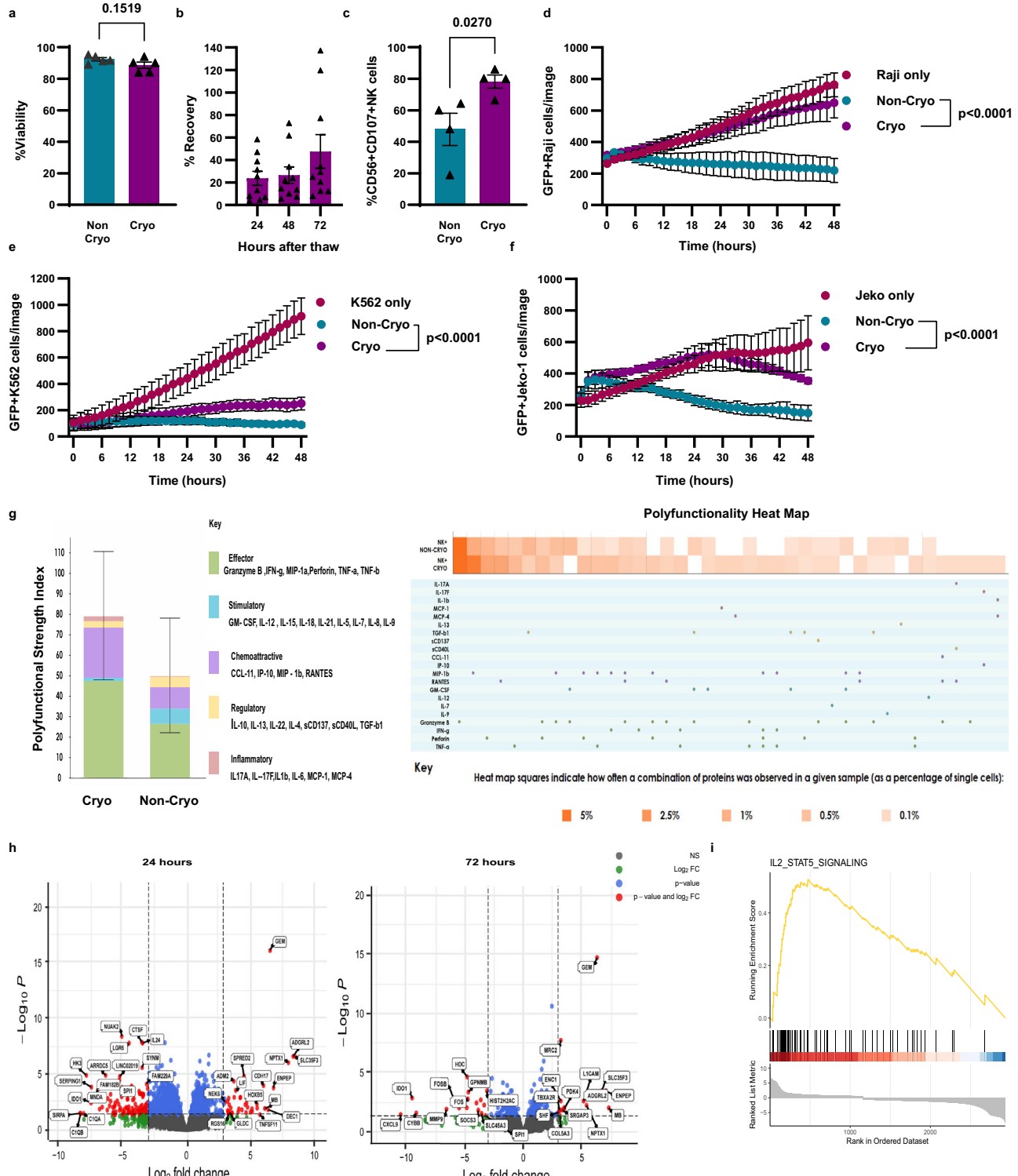

**Fig. 1 | Characterizing NK cell recovery, phenotype, and function after cryopreservation. a** NK cell viability after thawing assessed via flow cytometry (*n* = 5 healthy donors). **b** NK cell recovery 24, 48, and 72 h after thawing in comparison to starting NK cell number at time 0 h after thawing (*n* = 10 healthy donors). **c** CD107A degranulation assay before and after thawing. NK cells were cultured with K562 cells at 1:1 effector:target ratio for 3 h (*n* = 4 healthy donors). **d** NK cytotoxicity assay before and after cryopreservation with B2M KO Raji cells (*n* = 4 healthy donors). **e** NK cytotoxicity assay before and after cryopreservation with K562 cells (*n* = 4 healthy donors). **f** ADCC assay using anti-CD20 antibody and Jeko-1 cells (*n* = 3 healthy donors). **g** Single-cell secretome profile of cryopreserved and non-cryopreserved NK cells treated with R848 for 24 h. Heatmap showing individual cytokines that NK cell secreted (*n* = 3 healthy donors). **h** Bulk RNA volcano plot comparing before, 24 and 72 h after cryopreservation (*n* = 5 healthy donors). **i** GSEA of IL-2-STAT5 pathway. Normality test was used to determine the distribution of the data then parametric test *t* test was used for (**a, c**). Two-way RM ANOVA was used for (**d**–**f**). The Benjamini–Hochberg method was used to obtain *P* value shown in (**h**). A two-tailed test was used for (**a, c, h**). **g** Error bars are shown as mean SD. All other graphs are shown as mean ± SEM.

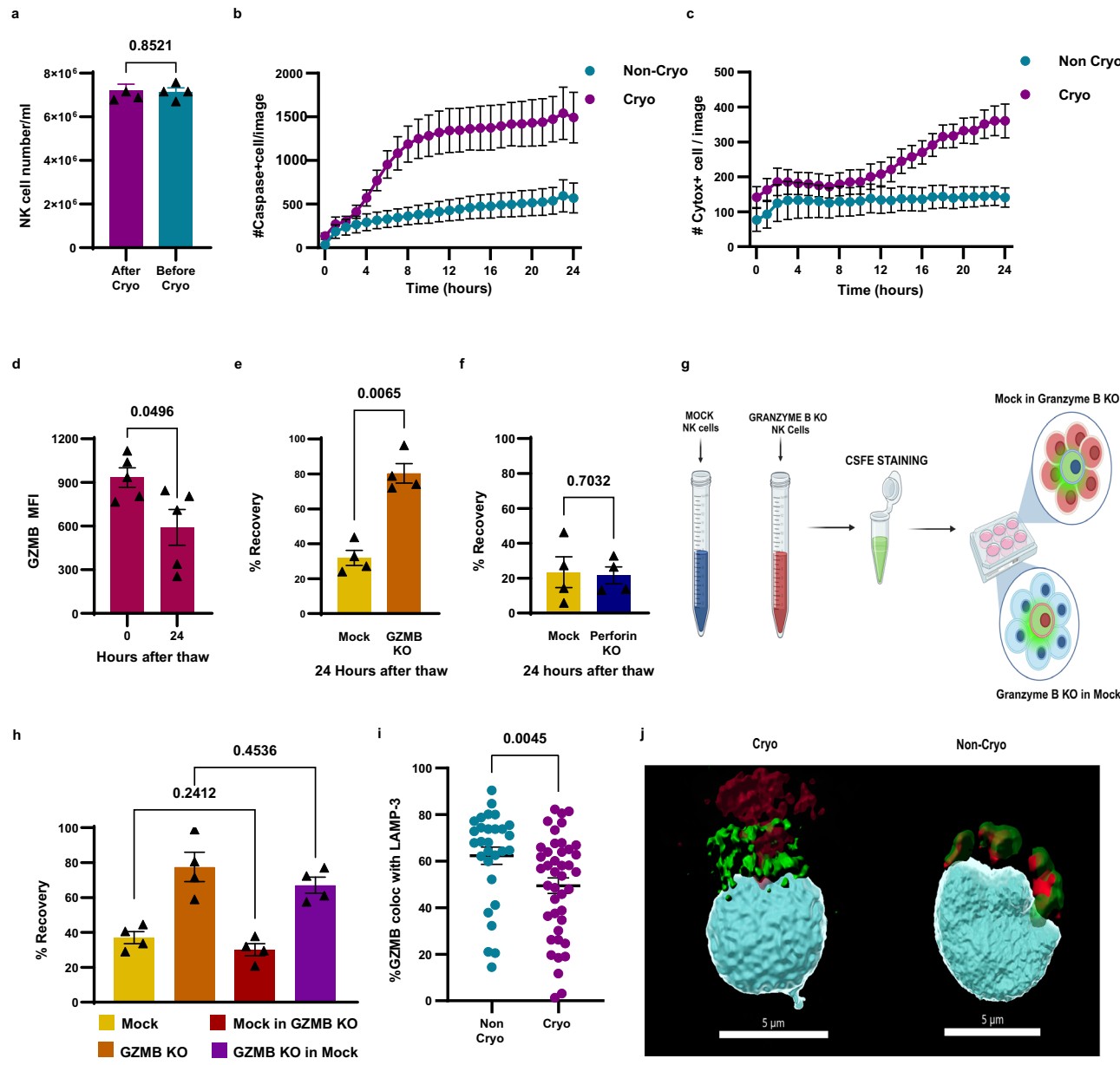

**Fig. 2 | GZMB leakage induces apoptosis in NK cells after cryopreservation.** **a** NK cell recovery immediately before and after cryopreservation ($n = 4$ healthy donors). **b** Incucyte live imagining using caspase-3/7 cleavable dye to indicate early apoptosis ($n = 4$ healthy donors). **c** Incucyte live imagining using Cytox NIR dye to indicate loss of plasma membrane integrity ($n = 4$ healthy donors). **d** granzyme B levels at time 0 and 24 h after thawing ($n = 5$ healthy donors). **e** GZMB knockout NK cells recovery after cryopreservation ($n = 4$ healthy donors). **f** Perforin knockout NK cells recovery after cryopreservation ($n = 4$ healthy donors). **g**, **h** GZMB and mock KO NK cells were cocultured at 1 ratio of 1:10 in both directions to determine the role of fratricide in NK cell death ($n = 4$ healthy donors). **i, j** Confocal microscopy imaging showing the extent of colocalization of GZMB and LAMP-3, a marker for cytotoxic granules in fresh versus cryopreserved NK cells (fresh $n = 29$, cryopreserved $n = 42$). Normality test was used to determine the distribution of the data then parametric test $t$ test was used for (**a, d, e, f, h**). nonparametric Mann–Whitney test was used for (**i**). A two-tailed test was used for (**a, d, e, f, h, i**). All graphs are shown as mean ± SEM.

Since cryopreserved NK cells have reduced cytotoxicity but increased CD107A expression, we measured intracellular levels of GZMB and found a reduction in intracellular GZMB levels 24 h post thaw (Fig. 2d), suggesting selective survival of GZMB$^{Lo}$ cells. However, this finding does not rule out the fratricide hypotheses because surviving NK cells may have recently degranulated (CD107A$^+$GZMB$^{Lo}$). To confirm the role of GZMB in the post-thaw apoptotic cell death pathway of NK cells, we conducted CRISPR-Cas9 editing to knockout (KO) *GZMB* and confirmed KO on the DNA and protein levels (Supplementary Fig. 2a). We next cryopreserved and thawed GZMB KO NK cells and observed a significant improvement in the numbers recovered post thaw (Fig. 2e).

To address whether NK cells are dying by fratricide or autolysis, we performed CRISPR-Cas9 editing on *PRF1* (perforin) and confirmed perforin KO on DNA and protein levels (Supplementary Fig. 2b). *PRF1* KO NK cells had similarly poor NK cell recovery after cryopreservation and thawing compared to mock KO NK cells (Fig. 2f), suggesting that GZMB is acting in a perforin-independent manner, favoring the autolysis hypothesis over fratricide. As there are reports that GZMB can kill target cells in a perforin-independent manner[27], we labeled *GZMB* KO cells with CFSE after thawing and cocultured them with an excess of unlabeled mock KO NK cells at an effector to target (E:T) ratio of 1:10, and also did the reverse by CFSE labeling mock KO NK cells and

coculturing them with an excess of unlabeled GZMB KO NK cells at the same E:T ratio (Fig. 2g). We hypothesized that if cells were dying of fratricide, we should see a reduction in the recovery of *GZMB* KO NK cells when cocultured with mock KO NK cells because the mock KO NK cells have intact GZMB. Similarly, mock KO NK cells should have a better recovery when cocultured with *GZMB* KO NK cells because fratricide would be largely dependent upon encountering other GZMB-intact mock KO NK cells, which are present at low proportions. The recovery of *GZMB* KO NK cells cocultured with an excess of mock KO NK cells and mock KO NK cells cocultured with an excess of *GZMB* KO NK cells was unaffected (Fig. 2h), confirming that fratricide does not play a meaningful role in NK cell apoptosis post thaw, which instead is potentially the result of autolysis.

Having identified that autolysis is the most likely cause of NK cell apoptosis post thaw, we examined whether GZMB leaks from cytotoxic vesicles post thaw using confocal microscopy via staining for GZMB and the vesicle marker LAMP-3[28,29]. We show decreased GZMB co-localized with LAMP-3 in cryopreserved NK cells compared to non-cryopreserved NK cells (Fig. 2i, j), supporting GZMB leakage from preformed cytotoxic vesicles.

Together, our data show that NK cells cryopreserved in GMP-compliant cryomedia containing 5 or 10% DMSO and using a controlled-rate freezer undergo programmed GZMB-dependent autolysis in the 24 h following thawing subsequent to the leakage of GZMB from cytotoxic vesicles. Neither necrosis nor fratricide appear to play a significant role.

## IL-15 + IL-18 pretreatment improves NK cell recovery and function after cryopreservation by upregulating antiapoptotic genes and temporarily reducing intracellular GZMB levels via degranulation

Having established that the loss of NK cell viability and function after cryopreservation is caused by GZMB-mediated autolysis rather than fratricide or necrotic cell death, we explored interventions that could be applied prior to cryopreservation to enhance NK cell recovery by mitigating the effects of GZMB leakage. First, we tested a small molecule reversible inhibitor of GZMB and an irreversible pan-caspase inhibitor, but neither compound improved the recovery of NK cells post thaw (Supplementary Fig. 3a). In both cases, it is possible the compounds did not adequately saturate the GZMB-caspase cell death pathway, especially given the likely protein turnover in the first 12–24 h post thaw, and for the reversible GZMB inhibitor, the compound was likely washed from the NK cells post thaw and/or diluted out in the fresh culture medium. We next screened cytokines, which represent a potentially ideal solution for both scientific and practical reasons. Indeed, cytokines, including IL-2[30], -15[31], and -18[32] have been shown to induce antiapoptotic pathways, with IL-18 in particular protecting NK cells from various inducers of apoptosis[32]. Moreover, in terms of practicality, GMP-grade cytokines are commercially available, readily integrated into NK cell manufacturing processes, and do not need to be present in the final cell product since their effects are indirect via receptor activation. Given prior data on IL-18 and the use of IL-12, -15, and -18 to produce cytokine-induced memory-like (CIML) NK[33] cells, we conducted a screen of the IL-1 family members (IL-1α, -1β, -18, -33, and -36), IL-12, and -15. We did not include IL-2 or IL-21 in the screen because our NK cells were already expanded with both cytokines and maintained in IL-2-supplemented media. We found that pretreating NK cells with IL-18 alone significantly improved their recovery post thaw, while IL-12 pretreatment reduced recovery (Fig. 3a, b). We tested pairwise and triple combinations of IL-12, -15, and -18 and found that NK cells pretreated with the IL-15 + IL-18 combination 24 h before cryopreservation had optimal recovery, while all combinations, including IL-12, showed no enhancement of recovery (Fig. 3a, b and Supplementary Fig 3c). Next, we treated NK cells with IL-15 + IL-18 for 0, 4, 8, 12, and 24 h before cryopreservation to establish the optimal

timepoint for cytokine pretreatment. We found a significant improvement in recovery by 4 h of exposure to IL-15 + IL-18, which did not improve significantly with up to 12 h of exposure, but was maximal at 24 h of exposure prior to cryopreservation, suggesting a mechanism that includes both rapid (<4 h) and slow (>12 h but ≤24 h) components (Supplementary Fig. 3d).

We were intrigued by the profound negative impact that IL-12 had on NK cell cryopreservation (Fig. 3b). To investigate, we stained NK cells treated with vehicle or IL-12 for GZMB, and we found that IL-12 upregulates GZMB significantly (Fig. 3c), which likely explains the negative impact of IL-12 on recovery after the thawing. Next, we explored the functionality of NK cells pretreated with IL-15 + IL-18 and found their cytotoxicity and ADCC were similar pre- and post-cryopreservation (Fig. 3d and Supplementary Fig. 3e, f). We had previously shown that cryopreserved NK cells have higher poly-functionality than non-cryopreserved NK cells (Fig. 1g), and this held true for IL-15 + IL-18-pretreated cryopreserved NK cells (Fig. 3e and Supplementary Fig. 3g), but is no longer associated with reduced cytotoxicity. We also measured caspase-3/7 activation in IL-15 + IL-18-treated cells post-thawing and found that IL-15 + IL-18-treated cells had caspase-3/7 activation signal above background but significantly lower than in vehicle-treated NK cells (Supplementary Fig. 3h).

IL-15 and IL-18 have multiple stimulatory effects on NK cells including the induction of degranulation[34] and the upregulation of antiapoptotic genes[31,32], both of which might both contribute to their cryoprotective effects. To explore degranulation, we examined CD107a staining and quantified secreted GZMB in the media and found that IL-15 + IL-18-treated NK cells reached peak degranulation after only 4 h, leading to significant elevations of GZMB in the media (Fig. 3f, g). We measured cell-associated GZMB levels by flow cytometry 24 and 72 h after IL-15 + IL-18 treatment and found a significant reduction in GZMB was seen at 24 but not 72 h after IL-15 + IL-18 (Fig. 3h), which is consistent with a temporary reduction by degranulation. To confirm that lower GZMB levels would not persist in IL-15 + IL-18-pretreated NK cells post thaw, we measured cell-associated GZMB levels at 0 and 24 h post thaw, as well as secreted GZMB levels at 24 h post thaw, (Supplementary Fig. 4a, b). We found increasing levels of cell-associated GZMB post thaw with the IL-15 + IL-18-pretreated NK cells retaining a higher secretory phenotype than vehicle-treated cells consistent with our single-cell secretome data (Fig. 3e). Based on these data, we hypothesized that reducing cell-associated GZMB before cryopreservation might improve recovery after thawing. To test this, we treated NK cells with DMSO or Cell Activation Cocktail (PMA + Ionomycin). We confirmed degranulation and reduction in GZMB levels in the PMA +Ionomycin treated cells and DMSO-treated cells (Supplementary Fig. 4c, d). Then these cells were cryopreserved and thawed. Recovery of the PMA/ Ionomycin treated cells trended towards improved recovery at levels similar to IL-15 + IL-18 after 4 h (Supplementary Figs. 3d and 4d) although it did not reach statistical significance (Supplementary Fig. 4d).

To determine if IL-15 + IL-18 treatment induces the upregulation in genes that may protect NK cells from GZMB-initiated apoptosis, we performed a bulk RNAseq analysis on NK cells that had been treated with IL-15 + IL-18 vs cells treated with vehicle control. RNAseq data showed upregulation of the BCL2 gene family, especially *BCL2L1* (Bcl-XL) (Fig. 3i). To confirm this result, we also used the NanoString nCounter assay using a panel with probes to detect genes involved in apoptosis and found that *BCL2L1* were one of the genes that are sig-nificantly upregulated (Fig. 3j). To confirm that BCL2L1 was upregu-lated on the protein level, we performed intracellular staining on NK cells treated with IL-15 + IL-18 and we detected a significant upregula-tion (Fig. 3k). Finally, we performed a CRISPR-Cas9 KO of *BCL2L1* and confirmed the KO on the DNA and protein level (Supplementary Fig. 4f) to determine whether BCL2L1 was necessary for the protective effects of IL-15 + IL-18. Our results showed profoundly diminished recovery of IL-15 + IL-18-pretreated BCL2L1 KO NK cells in comparison to control

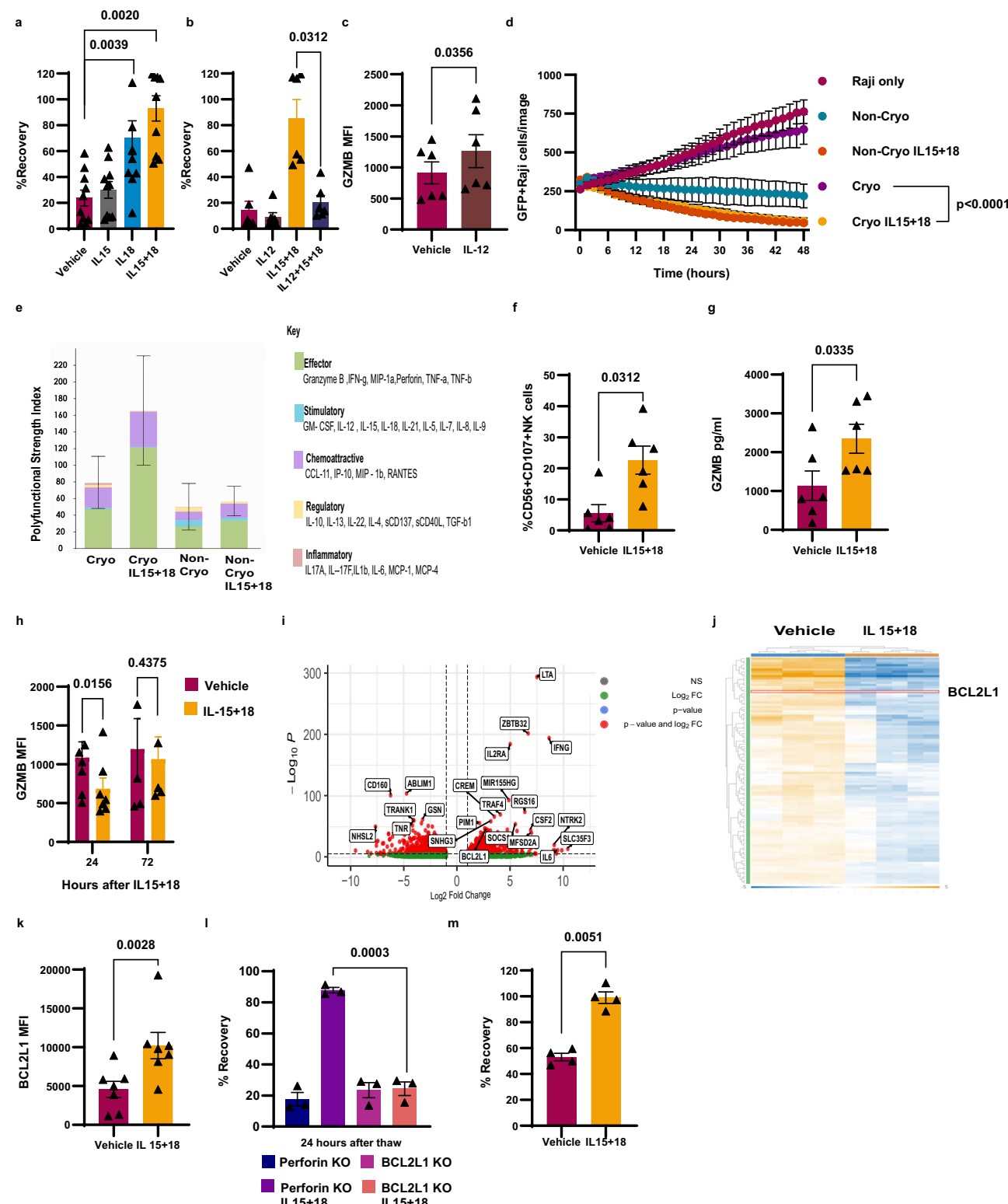

gene KO (perforin), highlighting this antiapoptotic factor is necessary to the mechanism for cryoprotection by IL-15 + IL-18 (Fig. 3i). Lastly, we used a commercial scale controlled-rate freezer (the CryoMed™ controlled-rate freezer) to mimic a GMP cryopreservation process and found that IL-15 + IL-18-treated NK cells showed 100% recovery of NK cells at 24 h post thaw (Fig. 3m).

Together our data demonstrate that pretreatment with a combination of IL-15 + IL-18 significantly and drastically improved NK cell recovery and function after cryopreservation. IL-15 + IL-18

pretreatment protects NK cells by upregulating antiapoptotic genes, specifically *BCL2L1*, which is necessary in rescuing NK cells from GZMB-initiated cell death and also by inducing degranulation, which temporarily reduces intracellular levels of GZMB and thus presumably the amount that may leak from preformed cytotoxic granules post thaw. Temporal evidence supports this two-part mechanism since some improvement of recovery is seen within 4 h of pretreatment, which is not improved upon until >12 h but ≤24 h of pretreatment, consistent with rapid degranulation followed by slower gene expression changes.

**Fig. 3 | IL-15 + IL-18 pretreatment improves NK cell recovery and function after cryopreservation by inducing GZMB release and upregulating antiapoptotic genes. a**, **b** Pretreating NK cells with cytokines 24 h before cryopreservation (n = 10 healthy donors (**a**) and n = 6 healthy donors for (**b**). **c** GZMB MFI 24 h after vehicle or IL-12 treatment (n = 7 healthy donors). **d** NK cell-mediated killing assay using IL-15 + IL-18-pretreated NK cell before and after cryopreservation (n = 4 healthy donors). **e** Single-cell secretome profile of IL-15 + IL-18-treated cryopreserved and fresh NK cells treated with R848 for 24 h (n = 3 healthy donors). **f** CD107A degranulation after IL-15 + IL-18 treatment (n = 6 healthy donors). **g** Granzyme B ELISA 24 h after IL-15 + IL-18 treatment (n = 6 healthy donors). **h** Comparing MFI of IL-15 + IL-18-treated NK cells at 24- and 72-h post-treatment (n = 7 healthy donors). **i** Volcano plot comparing IL-15 + IL-18-treated vs untreated

NK cells (n = 3 healthy donors). **j** Heatmap showing RNA levels using NanoString assay (n = 3 healthy donors). **k** MFI quantification of BCL2L1 after IL-15 + IL-18 treatment (n = 3 healthy donors). **l** Recovery of perforin KO and BCL2L1 KO NK cells with and without IL-15 + IL-18 pretreatment (n = 3 healthy donors). **m** IL-15 + IL-18-treated NK cell cryopreservation using GMP control rate freezer (n = 4 healthy donors). Normality test was used to determine the distribution of the data then parametric test t test was used for (**c**, **g**, **k**, **l**, **m**), nonparametric Wilcoxon matched pairs t test was used for (**a**, **b**, **f**, **h**). Two-way RM ANOVA was used for (**d**). The Benjamini−Hochberg method was used to obtain P value shown in (**i**, **j**). Two-tailed test was used for (**a**−**c**, **f**−**m**). **e** Error bars are shown as mean SD. All other graphs are shown as mean ± SEM.

## Deep immunoprofiling of the vehicle and IL-15 + IL-18-treated NK cells after cryopreservation using cytometry by time of flight (CyTOF) reveals distinct but reversible changes in NK cells phenotype

To explore the effects of IL-15 + IL-18 treatment on the phenotype of NK cells, we used a CyTOF panel comprised of 41 antibodies[35,36]. We thawed vehicle or IL-15 + IL-18- treated NK cells and stained them with viability dye and antibodies in a 10-day time course with analysis conducted on days 0, 1, 3, 7, and 10 post thaw. Live, intact, single-cells were then gated to resolve total CD45 + CD3−CD56 + NK cells (Supplementary Fig. 5a, b), and we identified a total of 15 unique clusters (Fig. 4a−c). At time 0 after thawing most vehicle-treated cells were in cluster 3 (Fig. 4d and Supplementary Fig. 5c). This cluster is CD56dim, CD69-, GZMB intermediate-high, whereas the majority of IL-15 + IL-18-treated NK cells fell into 5 clusters: 2, 9, 13, 14, and 15. All these clusters had intermediate-high levels of CD25, 4-1BB, CD69, and LAMP-1. These data show that IL-15 + IL-18 treatment activates NK cells in comparison to vehicle-treated cells, as anticipated from the biological functions of these cytokines. At day 1 and 3 after thawing vehicle control and IL-15 + IL-18-pretreated NK cells began to show a converging trend with increases in clusters 5, 6, 9, 10, and 13 in vehicle-treated NK cells, with the maintenance of clusters 9, 10, and 13 and increase in clusters, 5 and 6 in the IL-15 + IL-18-treated NK cells. Cluster 5 is NKp30 + , NKp44 +, TNFa+ and Ki-67 + , while cluster 6 has modest expression of a broad range of markers with high levels of KIR2DL3 and PD-1. By day 7 and 10 after thawing vehicle and IL-15 + IL-18-treated cells had almost identical cell distributions among the clusters indicating that IL-15 + IL-18 temporarily alters NK cells phenotype with NK cells returning to a phenotype in line with vehicle control-treated cells within a week of culture after thawing, maintaining NK cell heterogeneity.

## Cryopreserved IL-15 + IL-18-pretreated and NK cells are equipotent compared to non-cryopreserved NK cells in vivo

A key measure of the cryostability of cell therapy products is a comparison of non-cryopreserved versus cryopreserved cells in vivo. To examine this, we established a disseminated xenograft lymphoma model using *B2M* KO Raji cells, which are sensitive to NK cell-mediated killing independent of NK cell donor KIR haplotype[37]. NSG mice bearing disseminated *B2M* KO Raji cell lymphomas were treated with non-cryopreserved or cryopreserved NK cells IV with or without IL-15 + IL-18 pretreatment, and without any post-thaw recovery of NK cells in IL-2 media or dose adjustment (Fig. 5a). IL-15 + IL-18-pretreated cryopreserved NK cells achieved significantly improved tumor control, expansion in the peripheral blood, and survival compared to untreated cryopreserved NK cells, and in fact were equipotent as compared to non-cryopreserved untreated NK cells (Fig. 5b−d), indicating that the full potency of NK cells can be preserved by IL-15 + IL-18 treatment prior to cryopreservation.

## Discussion

Despite several decades of NK cell therapy trials, there remain scant reports comparing the activity and pharmacokinetics of non-

cryopreserved versus cryopreserved NK cells in patients[15]. The sole comparative report from the clinic showed that cryopreserved NK cells undergo profound cell death within 24 h post thaw, are not functional without pulsing with IL-2, and rarely expand in patients[13]. Unsurprisingly then, the use of non-cryopreserved NK cells remains at the forefront of innovation, for example in the landmark CD19 CAR-NK trial[7]. While there have been reports claiming to have improved NK cell cryopreservation via cryomedia or process optimization[12,17,19,20,38,39], we tested a high serum containing cryomedium, but were not able to reproduce these findings with EA NK cells in our own system. In our studies, we explored the mechanisms of NK cell death and observed that cryopreserved EA NK cells die between 12 h and 24 h post thaw due to autolysis subsequent to GZMB leakage from cytotoxic vesicles. The surviving NK cells had lower GZMB content and showed functional deficits matching prior data[13,14], although they were more polyfunctional upon R848 stimulation. Disparate claims in this field with respect to NK cell viability and function post thaw might be explained due to the timing and nature of the measurement of NK cell viability and function. We found no signs of caspase-3/7 activation, or vital dye uptake until 4 and 12 h post thaw, respectively, a time window in which viability assays and the commonly used 4 h $^{51}$Cr cytotoxicity assay are likely to be made on quality control vials of cryopreserved material. Overall, we found absolute viable cell count at 24 h post thaw was a better measure of NK cell recovery than the percentage of viable cells by flow cytometry due to the impact of gating strategies in flow cytometry analysis that exclude dead cells.

Having established the mechanism of NK cell death involved the leakage of GZMB from preformed cytotoxic granules, we sought interventions that would counter GZMB-mediated apoptosis. Unexpectedly, we were not able to rescue NK cells with inhibitors of GZMB or caspases despite using an irreversible small molecule inhibitor against the latter. This is likely due to protein turnover of GZMB and caspases within the first 24 h post thaw. In any case, we preferentially sought interventions that could be washed out prior to cryopreservation and thus would not lead to exposure in the patient or need reformulation of cryomedia. Cytokines fit these criteria as they can activate antiapoptotic pathways via interaction with their receptors and their effects last for some time[31,32]. In keeping with a prior report regarding the ability of IL-18 to protect NK cells from apoptosis resulting from chemical insult[32], IL-18 was able to significantly protect NK cells from cryopreservation-induced GZMB-mediated autolysis. In contrast, IL-12 negatively impacted NK cryostability and prevented the benefits of IL-18 when used in combination, which might have implications for the cryopreservation of CIML NK cells, which are produced with IL-12, -15, and IL-18 in combination and have become a subset of major interest due to their potency, persistence, and resistance to immune checkpoints[40]. IL-15, while not able to protect NK cells alone, enhanced the recovery and function of cryopreserved NK cells when used in combination with IL-18, achieving equipotency in a cytotoxicity assay, and in vivo in a model of disseminated B-cell lymphoma. While generally less functional than non-cryopreserved NK

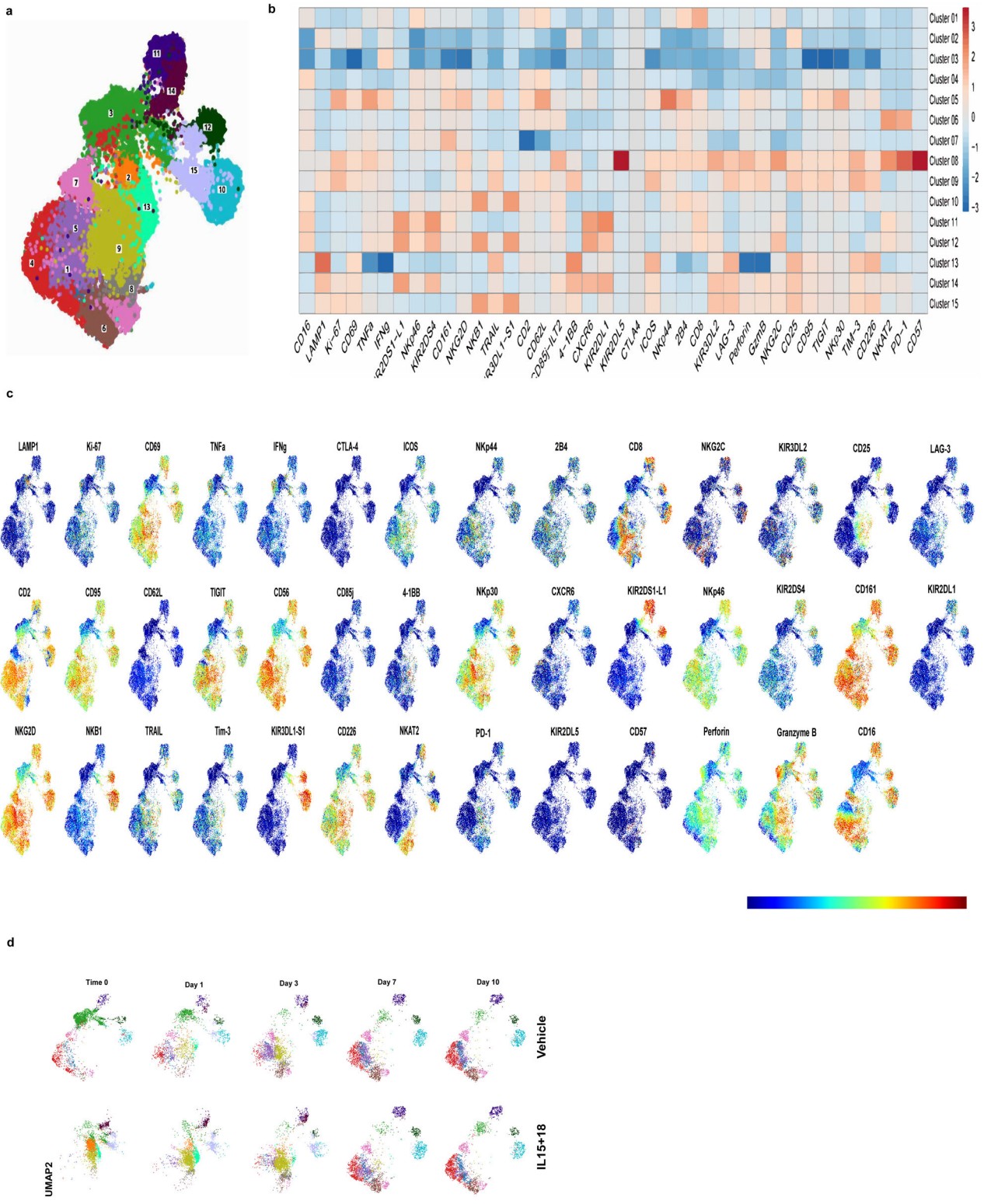

**Fig. 4 | Deep immunoprofiling of the vehicle and IL-15 + 18-treated NK cells after cryopreservation using cytometry by time of flight (CyTOF) reveals distinct changes in NK cell phenotypes.** **a** FlowSOM cluster map ($n = 3$ healthy donors). **b** individual marker expression in UMAP-1 ($n = 3$ healthy donors). **c** heatmap showing markers and expression levels in different clusters ($n = 3$ healthy donors). **d** UMAP showing NK cells distribution at different time points after thawing ($n = 3$ healthy donors).

cells, we observed that cryopreserved NK cells have a higher polyfunctional strength index, and this was also the case for those pretreated with IL-15 + IL-18. Phenotypically, cryopreserved IL-15 + IL-18-pretreated NK cells resting in IL-2 media showed phenotypic differences compared to vehicle control-treated NK cells due to the activating effects of these cytokines. However, a phenotypic convergence happens over several days showing marked convergence by day 3, and an indistinguishable phenotypic distribution by day 7

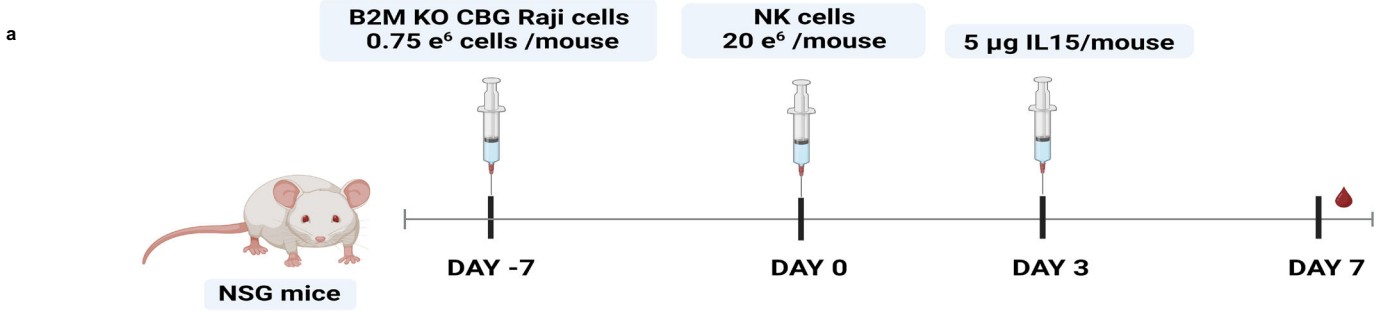

**Fig. 5 | IL-15 + IL-18 pretreatment enables equipotent in vivo activity of cryo-preserved and non-cryopreserved untreated NK cells. a** Schematic diagram for in vivo experiment. **b** Luminescent images of treated mice (*n* = 9 mice/group). **c** quantification of tumor burden (flux) (*n* = 9 mice/group). **d** NK cell number in the blood of treated mice (*n* = 9 mice/group). **e** Survival curve of NK treated mice (*n* = 9 mice/group) (**e**). Two-way ANOVA was used for (**c**). Normality test was used to determine the distribution of the data then parametric test *t* test was used for d. Simple survival analysis (Kaplan–Meier) with no data adjustment were made for (**e**). A two-tailed test was used for (**d**). All graphs are shown as mean ± SEM.

post thaw maintaining the heterogeneity of NK cells. Mechanistically, we found that IL-18 alone, and especially in combination with IL-15, enhances the expression of the antiapoptotic factor BCL2L1 at the RNA and protein levels, and that this factor is necessary for the improved cryostability of NK cells treated in this manner. Since IL-15 + IL-18 also induce degranulation and this is associated with a rapid and temporary reduction in intracellular levels of GZMB we posit this contributes to the cryoprotective mechanism, which displayed distinct rapid and slow onset, with similar modest increases in recovery of NK cells also achieved by treatment with an inducer of degranulation.

IL-15 and -18 are readily available as research- and GMP-grade reagents, and thus we anticipate that our findings with adult healthy donor EA NK cells may be broadly tested and applied if the method also benefits the cryostability of NK cells derived from umbilical cord blood, CD34+ stem cells, or induced pluripotent stem cells. Given that non-cryopreserved NK formulations continue to be applied at the cutting edge of the NK cell therapy field both in research labs and in the clinic, adoption of equipotent cryopreserved formulations may help accelerate the field.

## Methods

### Study design
The objective was to determine the mechanism of NK cell death after cryopreservation and identify an intervention to mitigate the loss of NK cell viability and function. We used genetic and immunological tools and assays together with microscopy techniques to test hypotheses related to NK cell function and viability. All experiments were repeated two or more times with NK cells from multiple healthy human donors. All reagents are listed below.

### Cells lines
K562 (CRL-3343), Raji (CLL-86), and Jeko-1 (CRL-3006) cell lines were obtained from the ATCC and engineered to express click beetle green luciferase (CBG). K562 feeder cells expressing 41BBL, CD40, and membrane-bound IL-21 (K-IL-21) were generated as previously described[41]. The identity of all cell lines was verified by STR profiling (ATCC), and they were confirmed negative for mycoplasma (Cambrex MycoAlert) following culture without antibiotics. Cells were cultured in RPMI 1640 (Thermo fisher #11875085) supplemented with 10% fetal bovine serum (FBS) (Seradigm #97068-086), and penicillin–streptomycin (Gibco# 15140122).

### NK cell isolation and expansion
Peripheral blood mononuclear cells (PBMCs) were obtained after written informed consent under the University of Pennsylvania Institutional Review Board (UP-IRB) (Protocol #705906, Federalwide Assurance # 00004028) approved protocol from healthy adult donors.NK cells were isolated and expanded for 15–20 days in G-Rex 6-well plates (Wilson Wolf # 80240M) via seeding with irradiated K-IL-21 feeder cells at an E:T ratio of 1:10 in X-VIVO 10 with Gent and Phenol Red (Lonza # BE04-380Q) supplemented with 10% human AB serum (Valley Biomedical #HP1022) and 100 U/ml of recombinant human IL-2 (Peprotech #200-02) (NK media). Fresh media was added to NK cells every 3 days as previously described[41].

### Cytokines and inhibitors
IL-15 at (Peprotech #200-15), IL-12 (Peprotech #200-12), IL-18 (Invivogen #rcyec-hil18), IL-1α (Peprotech #200-200-01A), IL-1β ((Peprotech #200-01B), IL-33 (Peprotech #200-33), IL-36 (Peprotech #200-36), Pan-Caspase Inhibitor Z VAD FMK (R&D Systems #FMK001), Brefeldin A Solution (1000X) (BioLegend #420601), Granzyme B Inhibitor II, Calbiochem (EMD Millipore #368055-1MG), Cell Activation Cocktail (BioLegend #423302) were used.

### Flow cytometry and CyTOF
All antibodies and buffers information can be found in Supplementary Table 1. For flow cytometry antibodies, we used 1:100 of antibody per 200,000 cells in 100 μl staining buffer. Cells were incubated at room temperature for 20 min then washed with staining buffer at 450×g for 5 min. Cells were then run on LSRFortessa flow cytometer. For CyTOF antibodies we used the manufacturer's recommended dilution 1:100 and instructions.

### Granzyme B ELISA
Human Granzyme B DuoSet ELISA was purchased from R&D Systems (#DY2906-05). ELISA kits were used according to the manufacturer's instructions.

### NK cell cryopreservation and thawing
NK cells were resuspended in NK media at $5 \times 10^6$ cells per ml. IL-15 and IL-18 were added at 50 and 250 ng/ml, respectively, while all other cytokines tested were added at 100 ng/ml 24 h before cryopreservation unless otherwise stated. NK cells were washed with media then resuspended in 1 ml of CryoStor CS5 or CryoStor CS10 (STEMCELL # 079331, 07930) before freezing to −80 °C in a CoolCell® cell freezer (VWR#75779-712) overnight followed by transfer to the vapor phase of liquid nitrogen for long-term storage. NK cells were thawed in a 37 °C water bath. When only a small piece of ice remained, cells were added dropwise to NK cell media at ratio of 1:10 volume of cells to media. Cells were centrifuged for 3 min at 400×g. Media was removed and cells were resuspended in NK cells media at $2.5–10 \times 10^6$ cells per ml.

### Degranulation assay
NK cells were cocultured with K562 cells for 3 h at 37 °C and then stained for flow cytometry analysis using a Fortessa flow cytometer. Percent degranulation was determined using the following gate CD3−CD56 + CD107a+ (% CD3−CD56 + CD107a+ cells/total CD3−CD56+ cells) *100.

### Single-cell cytokine release assay
NK cells were thawed and rested in NK media for 24 h prior to stimulation with 1 μg/ml R848 (Invivogen # tlrl-r848) for 16 h. NK cells were then stained with cell trace violet membrane dye and loaded into IsoPlexis single-cell human NK cell secretome chips, which were loaded into the IsoSpark Duo instrument and analyzed with IsoSpeak software according to the manufacturer's instructions.

### Microscopy
Non-cryopreserved or cryopreserved cells were fixed and stained with FITC-conjugated anti-CD63 and AF647-conjugated anti-Granzyme B antibodies. Cells were then washed twice and loaded onto EZ Double Cytofunnel (Fisher Scientific #A78710005) with glass slides. Cytofunnel was spun down to attach cells to the glass slides at 600 RPM for 10 min using mounting media ProLong Gold Antifade Reagent DAPI (ThermoFisher # P36931).

### Incucyte imaging for caspase-3/7, Cytox NIR, and killing assays
NK cells were thawed and immediately stained with Incucyte Caspase-3/7 Dye for Apoptosis-RED (Sartorius #4704), and Incucyte Cytotox Dye for Counting Dead Cells, NIR (Sartorius #4846) according to the manufacturer's instructions. NK cells were cocultured with GFP-expressing K562 or *B2M* KO Raji cells at 2:1, target-to-effector (T:E) ratio. Cells were incubated at 37 °C for 48 h and target cells fluorescence was measured every 1.5 h. ADCC assays using Jeko-1 cells as target cells were performed similarly to cytotoxicity assay but with the addition of anti-CD20 (Bio X Cell #SIM0008) at 1 μg/ml.

## Gene editing of NK cells and Raji cells

To generate *B2M* KO Raji cell line guide RNA targeting B2M was purchased form Synthego and used with SpyFi Cas9 Nucleases (Aldevron# 9214-5MG). gRNA was mixed with Cas9 for 20 min at RT to form RNP complex and then electroporated into Raji cells using a LONZA 4D system with SF buffer (LONZA#V4LC-2520). Guide RNAs targeting Granzyme B, Perforin and BCL2L1 were purchased form Synthego for RNP complexes were formed for 15 min at room temperature and then NK cells were electroporated using the BTX system at 500 μv for 700 μs then added to complete media at $2.5 \times 10^6$ cells/ml. Knockout efficiency was measured 3–4 days after electroporation by flow cytometry.all guid RNAs uase are listed in Supplementary Table 2. To confirm KO on the DNA level, we used these primers targeting Granzyme B, Perforin and BCL2L1 Supplementary Table 3.

## RNA sequencing and Nanostring nCounter assay

Total RNA was isolated from NK cells using the RNeasy Mini Kit (QIAGEN# 74104). Bulk RNA sequencing was performed at Novogen and Nanostring analysis was performed at the Wistar Institute genomic core. Bulk sequencing reads were aligned to the reference genome using the Hisat2 v2.0.5. Feature Counts v1.5.0-p3 was used to count the uniquely mapped gene. For differential expression analysis, the DESeq2R package (1.20.0) was used with *P*adj of 0.05 as described here[41]. For Nanostring, data were analyzed using the LBL-10025-02_PanCancer-Pathways panel and online software (www.rosalind.bio).

## Disseminated Raji lymphoma model

In all, 6–8-week-old male NOD.Cg-Prkdcscid Il2rgtm1Wjl/SzJ Stock (#005557) mice originally obtained from Jackson Laboratories were bred and maintained by the Stem Cell and Xenograft Core at the University of Pennsylvania in barrier mouse facility conditions. For all in vivo experiments, mice age 6–8 weeks were injected with 750,000 *B2M* KO Raji cell line expressing CBG. Day 7 after tumor injection mice were randomized into groups (9 mice/group) then injected with NK cells as described in Fig. 5a. Mice were injected with 5 μg of IL-15 (peprotech#200-15) on day 3 NK after injection. Mice were imaged every 3 days starting at day 0 after NK cell injection. Cheek bleed was performed at day 7 after NK cell injection and NK cells were counted using Trucount tubes (BD# 91-0786) according to the manufacturer's protocol. All mouse work was performed under an approved IACUC protocol(#804226) at the University of Pennsylvania. Approval was obtained from the University of Pennsylvania Institutional Animal Care and Use Committee, Office of Animal Welfare. Mice were euthanized if any of these conditions were met; loss of more than 20% of body weight, Bioluminescent Imaging (BLI) of the tumor reached $1 \times 10^{12}$ or paralysis in one or both limbs. Mice were euthanized using $CO_2$ chamber as approved by IACUC protocol. Mice were mentored 2–3 times a week.

## Statistical analyses

All statistical analyses were performed in GraphPad Prism (v9.0.1). The nature of the statistical test used is described in the figure legends, and corrections for multiple comparisons made where appropriate. A result ≤0.05 was considered statistically significant.

## Software and illustrations

For flow cytometry, the BD FACSDiva™ v8.0.1 Software was used. Incucyte® Base Analysis Software 2022B Rev3, Leica SP8 laser scanning confocal LAS X v3.5.7.23225. SoftMax Pro 6.2 plate reader software.

Flow Jo v10.8.1 GraphPad prism 10.0.2. Fiji ImageJ1.53t, Imaris image analysis, Ncounter data analysis Rosalind bio, Isospeak software. Living Image® Version 4.7 Software.

Illustration images in Figs. 2g and 5a were made using https://biorender.com/.

## Reporting summary

Further information on research design is available in the Nature Portfolio Reporting Summary linked to this article.

## Data availability

The RNA sequencing data have been deposited to NCBI database under these accession codes. Figure 1 data PRJNA1071276; Fig. 3 data PRJNA1071169. All other data will be available upon request. Source data are provided with this paper.

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

## Acknowledgements
The authors thank the Lanier and June lab members, especially Gina Young, Sofia Castelli, Divij Mathew, Eric Haas, Takuya Ohtani, and Aasha Gupta for providing protocols and critical discussion of this work. We thank Lynn Chen, Max Eldabbas, Miranda Wang and Emileigh Maddox of the Human Immunology Core (HIC) at the Perelman School of Medicine at the University of Pennsylvania for providing purified human NK cells. The authors also thank the UCSF Parnassus Flow Core (RRID:SCR_018206) for help and advice. The UPenn authors are supported by private funds. The UPenn HIC is supported in part by NIH P30 AI045008 and P30 CA016520. HIC RRID: SCR_022380. The USCF authors are supported by the Parker Institute for Cancer Immunotherapy (PICI) and NIH grants AI068129 and AI146581. The USCF Parnassus flow core is supported by the DRC Center Grant NIH P30 DK063720, Grant NIH P30 DK063720, and by the instrumentation grant NIH S10 1S10OD026940-01.

## Author contributions
A.B. and N.C.S. conceptualized the project. A.B., D.M., O.A., O.P., O.J., A.F., N.W.E., and L.C. developed the methods and performed the experiments. N.C.S., A.B., O.A.A., and L.L.L. wrote the manuscript. N.C.S., C.H.J., L.L.L., and O.A.A. reviewed and edited the manuscript. N.C.S., C.H.J., L.L.L., N.P., and J.S. supervised the project. C.H.J. and L.L.L. provided the funding.

## Competing interests
A.B. and N.C.S. have submitted a Patent application covering aspects of this work. C.H.J. has received grant support from Novartis, and has patents related to CAR-Therapy with royalties paid from Novartis to the University of Pennsylvania. C.H.J. is also a scientific co-founder and holds equity in BlueWhale Bio, Capstan Therapeutics and Tmunity Therapeutics. He serves on the board of AC Immune and is a scientific advisor Alaunos, BluesphereBio, Cabaletta, Carisma, Cartography, Cellares, Cellcarta, Celldex, Danaher, Decheng, ImmuneSensor, Poseida, Verismo, Viracta, and WIRB-Copernicus group. L.L.L. is an advisor to Catamaran, Cullinan Oncology, Dragonfly, DrenBio, Edity, GV20, IMIDomics, InnDura Therapeutics, Innovent, Nkarta, oNKo-innate, Obsidian Therapeutics, and SBI Biotech. N.C.S. holds equity in Tmunity Therapeutics and BlueWhale Bio. The remaining authors declare no competing interests.
