## [Peer Review File · Nature Communications]

Pretreatment With Interleukins 15 and 18 Rescues Natural Killer Cells from Granzyme B-Mediated Apoptosis After CryopreservationREVIEWER COMMENTS

Reviewer #1 (NK biology, cancer therapy.) (Remarks to the Author):

In the present work, Berjjs and coll study the mechanisms underlying reduced function/survival of cryopreserved NK cells and seek possible solutions to restore NK cell function after thawing. The authors provide evidence claiming that IL-15+IL-18 pretreatment before cryopreservation of GMP grade NK cell preparations maintains comparable survival and cytotoxicity with fresh NK cell preparations and that the overall phenotypic appearance of those cells after thawing is unchanged. Data on the efficacy of NK cells preparations in a mouse tumor model are provided to support the conclusions.

Comments:

Scope and interest of the work are elevated and the soluiton to intermediate NK cell apoptosis is promising.

Technical solutions to address the questions are fine, with a possible flaw in the methods used to ascertain actual target cell killing which are not investigated.

A thorough review of statistics is needed. for example, as indicated by Legend to Fig 1 where confusion of parametric and non parametric analysis is evident, and continues throughout the ms.

Parametric tests have been applied to nonparametric datasets.

Additionally, excessively downsized sample size does not provide a credible representation of the well-known variability of different patient samples and preparation replicates. Providing statistics on 2-3 samples is definitely not adequate to provide a plausible and beleivable estimate of inter-donor or inter-preparation variability

This applies to human experiments, while animal experiment data have a statistically accountable size.

Flow cytometric analysis of NK cells is based on a non-standard gating that includes in the analysis a relevant bias (lines 383-384) . this gate is likely to exclude a consistent fraction of adaptive NK cells which are NKp30. It is well established and also cited by the authors that pretreatment with IL12+15+18 induces and adaptive phenotype in CIML NK cells. Only a small fraction (10-40%) of adaptive NK cells are NKp30 positive. This analysis is therefore not suited to explore an induction of adaptive NK cells with IL15+IL-18 pretreatment before cryopreservation.

The authors present flow cytometric data at 0, 24, 48, 72 hrs after thawing of 1 sample. This is just a phenomenal analysis, and therefore the conclusions presented in the results and discussion are not warranted. In addition to the need of verifying multiple samples of multiple donors/preparations, and to gating bias, what happens to NK cell phenotypes after 7.-10 days of in vitro expansion/culture? Culture for 72hrs is sufficient to provide insight on early activation or subset loss, but not on the possible skewing effect of IL-15+IL-18 pretreatment on selective proliferation. Three days are not sufficient for addressing relevant outgrowth of a subset of NK cells, and are definitely not adapted on a patient time-scale.

Minor points.

Cytotoxicity is studied using assays that do not prove actual target cell lysis by NK cells. Restoring Cytotoxicity needs proof of target cell lysis.

Some supplementary data should be presented in the figures. The IL-12 data that are discussed in the discussion section should appear in figures, as they are not indeed ancillary data but rather central to the discussion on CIML.

Reviewer #2 (NK/ILC biology, immune signaling.) (Remarks to the Author):

NK cell-based therapies are promising approaches for immunotherapies. However, cryopreservation presents a major hurdle for potential NK cell-based off-the-shelf products as recovery and viability of cryopreserved NK cells is low after thawing. Here the authors identify granzyme B leakage from cytotoxic granules as a potential mechanism for this effect. They convincingly show that deletion of granzyme B but not perforin can increase NK cell recovery 24h post-thaw. They also demonstrate that pre-treatment of NK cells with IL-15 and IL-18 before cryopreservation can increase recovery and functionality of NK cells post-thaw. However, it is unclear if both mechanisms are connected. The authors show that IL-15/18 treatment can lower granzyme B levels by inducing degranulation. As IL-15/18 pre-treatment also has beneficial effects on non-frozen NK cells (figure 3b and figure 5) and as it also induces Bcl-2L expression, it is unclear if the effect of IL-15/18 is indeed connected to granzyme B leakage. Therefore, it is important to show that the lower granzyme B levels of IL15/18 pre-treated NK cells are sufficient to prevent granzyme B leakage (as shown in figure 2j) and cell death (as shown in figure 2b,c) in order to support the claim of the manuscript title.

Minor points:

1. Line 197: The sentence seems to be incomplete.
2. Figure 1 g, h: The RNAseq data are not really discussed by the authors, and to me it is unclear if they provide any relevant information for the manuscript. The authors could compare them to the RNAseq data of the IL-15/18 treated NK cells to increase their relevance or simply delete them.
3. Figure 1: The legend lacks the description of the 'regulators' cytokines.
4. Figure 1 and 3: I find it interesting that the cryopreserved NK cells almost completely lack their regulatory secretome (no secretion of TGF-b1 or GMCSF) and this cannot be rescued by IL15/18 treatment. The authors may want to comment on this in the discussion.

Reviewer #3 (NK biology, neonatal immunity.) (Remarks to the Author):

This is an extremely interesting and very timely study that examines the underlying mechanism for the poor efficacy of cryopreservation of primary human NK cells which is a significant hindrance to NK cell immunotherapy therapeutics.

The major finding of the paper is that GZMB levels are critical to determining the efficacy of cryopreservation. A critical piece of supporting data is the ELISA in Figure 3e. It appears that is a "n=1" result and the figure shows data from 3 technical replicates. Is that correct?

The ELISA in Figure 3e examines GZMB levels post-treatment with IL-15 & IL-18. It is critical to see the same analysis after cryopreservation, post-thaw. Flow cytometry is used to determine intracellular GZMB levels in Figure 2d in untreated NK cells after thaw. Again this appears to be a "n=1" event. Similarly this method could be used to determine GZMB levels in IL-15 & IL-18 treated cells post thaw. It is not shown in the paper.

The KO data presented in the paper is used to strongly support the central hypothesis and underlying mechanism regulating apoptosis. It is customary to present an estimate of the targeting efficiency, particularly as the authors use a sgRNA methodology that recommends indel identification by DNA sequencing of the targeted cells and provides software to do this precise analysis. The flow cytometry characterisation of the GZMB KO and Perforin KO are given in Supplementary Figure 2. The plot shown in SSC versus GZMB/Perforin, why not CD56 versus GZMB/perforin if we are looking at targeted human NK cells? No indication is given of the efficiency of the BCL2L1 KO.

To aid reproducibility, the authors needs to indicate how long the cells spent in liquid nitrogen prior to thawing for experiments. In addition, it has long been known that cell concentration is a critical parameter for cryopreservation, how did the authors arrive at a freezing cell concentration of up to 1×10^7 cells/ml.

The Figures 3g and 3h are used to indicate that BCL2L1 is significantly upregulated. This is not justified in any clear fashion, upregulated to compared to what? One BCL-2 family member has been very clearly identified to block NK cell apoptosis and respond to IL-15 (Nat Immunol 2007 8:856-863) and that is MCL-1, the authors should at least comment on this and indicate that it does not appear significant from their analysis.

Minor comments

1. Please provide a rationale for the use of the single cell secretome of NK cells treated with TLR7/8 agonist
2. The authors need to make clear why Figure 4 is not a Supplementary Figure, it adds very little to the paper
3. I'm not clear why n=9 in Figure 5c but we are only shown n=5 in Figure 5b
4. The presentation of the Volcano plots shows multi-coloured area, no indication is given what these different colours signify.
5. Could the authors please indicate why they use K562 cells as targets in degranulation assays after the NK cells have been expanded following co-culture with K562 cells?
6. The Figure Legends could be much clearer.
7. The text has a lot of typos.

Submitted Manuscript:**Pretreatment With Interleukins 15 and 18 Rescues Natural Killer Cells from Granzyme B-Mediated Apoptosis After Cryopreservation****Response to Reviewer #1:**

Reviewer #1's summary of manuscript: In the present work, Berjis and coll study the mechanisms underlying reduced function/survival of cryopreserved NK cells and seek possible solutions to restore NK cell function after thawing. The authors provide evidence claiming that IL-15+IL-18 pretreatment before cryopreservation of GMP grade NK cell preparations maintains comparable survival and cytotoxicity with fresh NK cell preparations and that the overall phenotypic appearance of those cells after thawing is unchanged. Data on the efficacy of NK cells preparations in a mouse tumor model are provided to support the conclusions. Scope and interest of the work are elevated and the solution to intermediate NK cell apoptosis is promising.

We thank you for this kind and insightful summary.

Major Comment 1: *Technical solutions to address the questions are fine, with a possible flaw in the methods used to ascertain actual target cell killing which are not investigated.*

Although luciferase based killing assays are commonly used to assess NK target killing (PMID: 29317736). We do believe you have a valid point as we have measured killing indirectly throughout the submitted version of our manuscript. To address this concern, we replaced all of our in vitro killing assays with live imaging based killing assay (Incucyte). GFP+ Raji, K562 and Jeko-1 target cells were imaged every 1.5 hours for 48 hours with and without NK cells. Please see Figure 1, 3 and S3 for the new data where cytotoxicity is directly measured.

Major Comment 2: *A thorough review of statistics is needed. for example, as indicated by Legend to Fig 1 where confusion of parametric and non parametric analysis is evident, and continues throughout the ms. Parametric tests have been applied to nonparametric datasets.*

We thank you for this key comment to strengthen the manuscript. In the revised version we have first applied normality testing and then used parametric or non-parametric tests as appropriate to the distribution of the data.

Major Comment 3: *Additionally, excessively downsized sample size does not provide a credible representation of the well-known variability of different patient samples and preparation replicates. Providing statistics on 2-3 samples is definitely not adequate to provide a plausible and believable estimate of inter-donor or inter-preparation variability. This applies to human experiments, while animal experiment data have a statistically accountable size.*

We concur with you. In our original submission there were instances showing representative donors. We have made sure that in the revision all donors have been shown. This applies to Figure 1 c - f, Figure 2 a – f and h, Figure 3 b – f, i, and j, Figure S 1c, Figure S3 a, b , e, and f. In all cases no fewer than 3 biological replicates (healthy donor NK cells) are shown, and often the n number is higher as stated in figure legends.

Major Comment 4: *Flow cytometric analysis of NK cells is based on a non-standard gating that includes in the analysis a relevant bias (lines 383-384) . this gate is likely to exclude a consistent fraction of adaptive NK cells which are NKp30. It is well established and also cited by the authors that pretreatment with IL12+15+18 induces an adaptive phenotype in CIML NK cells. Only a small fraction (10-40%) of adaptive NK cells are NKp30 positive. This analysis is therefore not suited to explore an induction of adaptive NK cells with IL15+IL-18 pretreatment before cryopreservation.*

Thank you for raising this important point. We repeated the phenotypic analysis during the revision to address this concern and to address a related concern that the analysis period was too short. The new dataset uses CyTOF rather than Spectral Cytometry, with NK cells gated as CD45+CD3-CD56+ cells after standard CyTOF cleanup gates. Please see Figure 4 and S5 in the revised manuscript.

Major Comment 5: *The authors present flow cytometric data at 0, 24, 48, 72 hrs after thawing of 1 sample. This is just a phenomenal analysis, and therefore the conclusions presented in the results and discussion are not warranted. In addition to the need of verifying multiple samples of multiple donors/preparations, and to gating bias, what happens to NK cell phenotypes after 7.-10 days of in vitro expansion/culture? Culture for 72hrs is sufficient to provide insight on early activation or subset loss, but not on the possible skewing effect of IL-15+IL-18 pretreatment on selective proliferation. Three days are not sufficient for addressing relevant outgrowth of a subset of NK cells, and are definitely not adapted on a patient time-scale.*

To address your concerns during the revision we ran NK cells from n = 3 donors at time = 0, 1, 3, 7 and 10 days using CyTOF. Please see revised figure 4. We found IL-15+IL-18 pretreatment did result in baseline differences in NK phenotype particularly with respect to activation markers as may be anticipated, however the differences were minimal by day 3, and absent at days 7 and 10. This sets the IL-15+IL-18 pretreatment apart from reports of CIML programming with IL-12+IL-15+IL-18, where durable skewing towards memory subsets occurs.

Minor Comment 1: *Cytotoxicity is studied using assays that do not prove actual target cell lysis by NK cells. Restoring Cytotoxicity needs proof of target cell lysis.*

We concur with you and have replaced the cytotoxicity data with that from a live cell imaging (Incucyte) platform.

Minor Comment 2: *Some supplementary data should be presented in the figures. The IL-12 data that are discussed in the discussin section should appear in figures, as they are not indeed ancillary data but rather central to the discussion on CIML.*

We agree that these data could be of interest broadly in the NK cell research community. We have therefore highlighted in the revised Figure 3 (part b, c) the effect of IL-12 on NK cell recovery and also found it out the IL-12 increases Granzyme B levels in NK cells which could explain the poor recovery of IL-12 treated NK cells after cryopreservation.

Response to Reviewer #2:

Reviewer #2's summary of manuscript: NK cell-based therapies are promising approaches for immunotherapies. However, cryopreservation presents a major hurdle for potential NK cell-based off-the-shelf products as recovery and viability of cryopreserved NK cells is low after thawing. Here the authors identify granzyme B leakage from cytotoxic granules as a potential mechanism for this effect. They convincingly show that deletion of granzyme B but not perforin can increase NK cell recovery 24h post-thaw. They also demonstrate that pre-treatment of NK cells with IL-15 and IL-18 before cryopreservation can increase recovery and functionality of NK cells post-thaw.

We thank you for this summary of our work and are glad these points came across clearly in the submitted version.

Major Comment 1: *However, it is unclear if both mechanisms are connected. The authors show that IL-15/18 treatment can lower granzyme B levels by inducing degranulation. As IL-15/18 pre-treatment also has beneficial effects on non-frozen NK cells (figure 3b and figure 5) and as it also induces Bcl-2L expression, it is unclear if the effect of IL-15/18 is indeed connected to granzyme B leakage. Therefore, it is important to show that the lower granzyme B levels of IL15/18 pre-treated NK cells are sufficient to prevent granzyme B leakage (as shown in figure 2j) and cell death (as shown in figure 2b,c) in order to support the claim of the manuscript title.*

Thank you for this comment. We concur that we needed to do more to adequately demonstrate the respective contributions of the two proposed mechanisms that are both induced by IL-15+IL-18.

Our BCL2L1 CRISPR KO data lead us to believe that this is the key mechanism since intact BCL2L1 expression was necessary for IL-15+IL-18-mediated enhancement of recovery. We have made sure to highlight this as the dominant mechanism in the revised version.

We wish to clarify that we do not believe IL-15+IL-18 prevent the leakage of GZMB by reducing its intracellular levels, only that such leakage would be of lower magnitude if the total intracellular levels are lower, and thereby lead to lower magnitude activation of the Caspase 3/7 pathway, and perhaps be easier to counter by upregulating BCL2L1. Unfortunately, we are not in possession of assays suited to directly quantify non-cytotoxic granule associated GZMB of sufficient sensitivity to definitely investigate this point.

Your comment led us to redouble our efforts to design additional assays. We thought the best assay would be one where we could give degranulation inhibitors to NK cells pretreated with IL-15+IL-18 to show the extent to which recovery might be affected when degranulation is blocked. To this end calcium release-activated calcium channel protein 1 (ORAI1) has been shown to regulated NK cell degranulation and mutations in this gene leads to reduced NK cell function (PMID: 21300876). We made attempts with six sgRNAs to knockout ORAI1 by CRISPR gene editing but were unsuccessful as we could not see reduction of expression on the protein level. Next, we identified that DPB162-AE is an inhibitor of ORAI1 (PMID: 28196740). Unfortunately, this small molecule is not available from commercial sources, and while we approached three groups who have published manuscripts using the compound, we did not receive any responses. AnCoA4 is a commercially available ORAI1 inhibitor, which may be less selective and potent than DPB162-AE given the preference for the latter in the literature. We

obtained and tested AnCoA4 at multiple concentrations and time points but we did not see a difference in degranulation when we gave cells IL-15+IL-18 (please see data below).

With these avenues proving unfruitful, we finally decided to come at the issue from the other direction by treating NK cells with an inducer of degranulation (PMA+Ionomycin) instead of IL-15+IL-18 and for a short time period (4h) that would not be conducive of global changes in gene expression. These new data (See Suppl. Fig 4 c, d, e) showed that PMA+Ionomycin induced degranulation, reduced cell-associated GZMB levels and trended towards improving recovery to a degree that was meaningful by average % change, but did not reach significance due to one outlier donor. These data also fit well with our prior data that a cryoprotective effect of IL-15+IL-18 can be observed by 4h but doesn't improve markedly until >12 and ≤ 24h incubation potentially indicative of two stage mechanism involving (1) dumping of GZMB by degranulation (rapid), followed by (2) upregulation of BCL2L1 expression subsequent to an IL-15+IL-18-induced gene expression changes (slower).

While not a perfect substitution for the degranulation inhibition concept that we could not get to work in the time available, we hope you agree that the PMA + Ionomycin data support the potential of GZMB dumping by degranulation to be involved in the cryoprotective effect of IL-15+IL-18, with the main mechanism being upregulation of BCL2L1 to counter the GZMB-initiated apoptotic signal.

To address your latter point, we tried to do use the same imaging-based assay to measure caspase 3/7 activity after thawing in IL-15+IL-18 treated cells. This assay depends on being able to segment individual cells then fluorescent positive cells can be counted. Unfortunately, it was impossible to segment IL-15+IL-18 treated cells due to high clustering please see images below. That being said, we still wanted to address you comment so we used another caspase 3/7 assay (Promega G8091 Caspase-Glo® 3/7 Assay System). This assay uses a similar concept to the assay we have in figure 2 but instead of fluorescence it uses luminescence which does not require cells segmentation. We included the data in the paper Figure S 4h.

Minor Comment 1: *Line 197: The sentence seems to be incomplete.*

Thank you! This seems to have been an error from the editing process. We have corrected it.

Minor Comment 2: *Figure 1 g, h: The RNAseq data are not really discussed by the authors, and to me it is unclear if they provide any relevant information for the manuscript. The authors could compare them to the RNAseq data of the IL-15/18 treated NK cells to increase their relevance or simply delete them.*

Thank you for this comment! IL-15 and IL-18 are both potent stimulators of NK cells and of course did not evolve to have any role in cryopreservation which is our key interest. We anticipated that many DEGs would be apparent but that the vast majority of these would have nothing to do with the potential cryoprotective effects of IL-15+IL-18. We include these data largely for context of how the antiapoptotic genes are regulated versus general changes.

Minor Comment 3: *Figure 1: The legend lacks the description of the 'regulators' cytokines.*

Thank you for pointing this out. We have included the keys next to the figure.

Minor Comment 4: *Figure 1 and 3: I find it interesting that the cryopreserved NK cells almost completely lack their regulatory secretome (no secretion of TGF- β 1 or GM-CSF) and this cannot be rescued by IL15/18 treatment. The authors may want to comment on this in the discussion.*

When we combined data from multiple donors together for the revision this observation no longer held.

Response to Reviewer #3:

Reviewer #3's summary of manuscript: *This is an extremely interesting and very timely study that examines the underlying mechanism for the poor efficacy of cryopreservation of primary human NK cells which is a significant hindrance to NK cell immunotherapy therapeutics.*

We are grateful to you for this kind summary of our manuscript.

Major Comment 1: *The major finding of the paper is that GZMB levels are critical to determining the efficacy of cryopreservation. A critical piece of supporting data is the ELISA in Figure 3e. It appears that is a "n=1" result and the figure shows data from 3 technical replicates. Is that correct? The ELISA in Figure 3e examines GZMB levels post-treatment with IL-15 & IL-18. It is critical to see the same analysis after cryopreservation, post-thaw.*

Thank you for highlighting this. The original submission used one representative donor, but in the revised version we combined data from all normal donors into one graph. Please see figure 3 g.

We also performed a GZMB ELISA post thawing and included the data in Figure S 4 b.

Major Comment 2: *Flow cytometry is used to determine intracellular GZMB levels in Figure 2d in untreated NK cells after thaw. Again this appears to be a "n=1" event. Similarly this method could be used to determine GZMB levels in IL-15 & IL-18 treated cells post thaw. It is not shown in the paper.*

We concur and have now combined data for all donors/biological replicates. Please see the revised figure 2 d. We also performed GZMB staining post thawing and included the data in Figure S 4 a.

Major Comment 3: *The KO data presented in the paper is used to strongly support the central hypothesis and underlying mechanism regulating apoptosis. It is customary to present an estimate of the targeting efficiency, particularly as the authors use a sgRNA methodology that recommends indel identification by DNA sequencing of the targeted cells and provides software to do this precise analysis.*

Thank you for this comment. We agree and have analyzed the KO efficiency on DNA level and included indels for all knockouts. Please see Figure S2 a and b, and Figure S4 f.

Major Comment 4: *The flow cytometry characterisation of the GZMB KO and Perforin KO are given in Supplementary Figure 2. The plot shown in SSC versus GZMB/Perforin, why not CD56 versus GZMB/perforin if we are looking at targeted human NK cells? No indication is given of the efficiency of the BCL2L1 KO.*

To address this point, we have changed gating to show CD56 instead of SSC. Please see figure S2 a and b and Figure S4 f.

Major Comment 5: *To aid reproducibility, the authors needs to indicate how long the cells spent in liquid nitrogen prior to thawing for experiments. In addition, it has long been known that cell concentration is a critical parameter for cryopreservation, how did the authors arrive at a freezing cell concentration of up to 1×10^7 cells/ml.*

All NK cells spent a minimum of 72h in LN2, and a maximum of 2 months. We have not conducted long-term stability studies, but cellular products stored in LN2 have been known to be viable and essentially unaltered for decades. We wondered if the Reviewer may be concerned about the minimum time under these conditions, which were driven in our case by certain studies where non cryopreserved and cryopreserved material from the same donors must be compared without excessive passage of the non-cryopreserved NK cells. Cryoinjury is driven by the formation and subsequent dissolution of ice crystals, and thus we posit any damage is done during the controlled rate freeze to -80°C rather than during the more rapid temperature change down to the vapor phase of LN2 which occurs after the cryovials are removed from the controlled-rate freezer, placed in a rack and immediately lowered into the LN2 tank. Thawing is always rapid in a 37°C water bath.

We agree with the Reviewer that density is critical, and we based our admittedly narrow range (1 and 2×10^7 mL were tested and performed equally) on those used for CAR-T and CAR-NK product. For infusion of Kymriah[®] for example, 1-3 bags of 10-50mL are infused at 10-20mL per minute for up to 2.5×10^8 CAR-T cells total dose. Per Prof. Rezvani's recent CAR-NK trial (PMID: 38238616), a flat dose of 8×10^8 CAR-NK cells was used, which would require 40 - 80mL at $2 \times 10^7/\text{mL}$ at our formulations, which is in line with typically infused volumes for cellular therapies such as Kymriah[®]. We therefore adopted the 1- 2×10^7 range as being practical for the purposes of clinical infusions and thus relevant to clinical translation, but we accept that the eventual clinical translation process may require more exploration of NK cell densities.

Major Comment 6: *The Figures 3g and 3h are used to indicate that BCL2L1 is significantly upregulated. This is not justified in any clear fashion, upregulated to compared to what?*

Thanks for catching this. We have changed the text in figure 3 to be clearer. We are comparing vehicle treated cells vs IL-15+IL18 treated cells. IL-15+IL-18 significantly upregulated BCL2L1 in comparison to vehicle treated NK cells.

Major Comment 7: *One BCL-2 family member has been very clearly identified to block NK cell apoptosis and respond to IL-15 (Nat Immunol 2007 8:856-863) and that is MCL-1, the authors should at least comment on this and indicate that it does not appear significant from their analysis.*

Thanks for raising this intriguing point. We looked at our RNA seq data and MCL-1 and found it was not significantly altered, although it is possible significance is lost in the noise of RNAseq data.

Minor Comment 1: *Please provide a rationale for the use of the single cell secretome of NK cells treated with TLR7/8 agonist*

The use of R848 is recommended by IsoPlexis (now Bruker Cellular Analysis) as the stimulant for NK cells. We have previously found it to more potently trigger cytokine production than stimulation with

K562 target cells (PMID: 37649085), and thus provide a more detailed view of the cytokine secretion potential of the NK cells.

Minor Comment 2: *The authors need to make clear why Figure 4 is not a Supplementary Figure, it adds very little to the paper*

During the revision we replaced the original data with CyTOF and have rewritten this section. While programming of NK cells with IL-12, -15 and -18 to induce the CIML phenotype is known to be durable and significant we wanted to show whether IL-15+IL-18 treatment led to major changes and found that only short-term differences were observed. At high resolution then our pre-treatment only temporarily alters the NK phenotype, and this may be important for labs that are studying specific kinds of NK cell expansion etc and do not wish to push their NK cells into a permanent phenotypic change.

Minor Comment 3: *I'm not clear why n=9 in Figure 5c but we are only shown n=5 in Figure 5b*

We have now included all 9 in figure 5c.

Minor Comment 4: *The presentation of the Volcano plots shows multi-coloured area, no indication is given what these different colours signify.*

Thanks for catching this. We have included the figure key next the figures.

Minor Comment 5: *Could the authors please indicate why they use K562 cells as targets in degranulation assays after the NK cells have been expanded following co-culture with K562 cells?*

The purpose of this assay is only to show whether differences exist between the conditions and K562 cells being HLA negative are potent stimulators of NK cells regardless of KIR type. In our expansion process, the K562 feeder cells are irradiated and are depleted by the combined action of the NK cells and radiation-induced apoptosis within 48-72h of coculture. The NKs are expanded until day 15 at which point, they are fully rested and show marked response upon restimulation with K562 cells.

Minor Comment 6: *The Figure Legends could be much clearer.*

Thank you for this comment. We have made improvements to all figure legends.

Minor Comment 7: *The text has a lot of typos.*

Thank you for highlighting this. We have combed the revised manuscript for typos with spell check and careful review.

REVIEWERS' COMMENTS

Reviewer #1 (Remarks to the Author):

The authors addressed all comments raised and updated/improved the manuscript and added new data which are in line with the conclusions.

Together with comments raised and similarly addressed by the other reviewers, the manuscript has been substantially improved, with renewed additional work that renders now the work of high interest and relevance.

Reviewer #2 (Remarks to the Author):

The authors have sufficiently addressed all my concerns in their revised version. I have no more comments.

Reviewer #3 (Remarks to the Author):

The authors have done a thorough job addressing my comments. I think on balance the paper is now ready to publish and to make this new and important data available to the field.